# Examination of the cognitive function of Japanese community-dwelling older adults in a class for preventing cognitive decline during the COVID-19 pandemic

**Minoru Kouzuki** [ORCID] *, **Shota Furukawa, Keisuke Mitani, Katsuya Urakami**

Department of Biological Regulation, School of Health Science, Faculty of Medicine, Tottori University, Tottori, Japan

* kouzuki@tottori-u.ac.jp

**Data Availability Statement:** The data sets used and/or analyzed during the current study are available from the corresponding author or the

## Abstract

We examined the changes in cognitive function due to restrictions in daily life during the coronavirus disease 2019 (COVID-19) pandemic in community-dwelling older adults with mild cognitive decline. This was a retrospective, case-control study. The participants include 88 older adults with mild cognitive decline (mean age = 81.0 [standard deviation = 6.5] years) who participated in a class designed to help prevent cognitive decline. This class was suspended from early-March to end of May 2020 to prevent the spread of COVID-19, and resumed in June 2020. We collected demographic and cognitive function test data (Touch Panel-type Dementia Assessment Scale [TDAS]) before and after class suspension and questionnaire data on their lifestyle and thoughts during the suspension. Change in TDAS scores from before and after the suspension was used to divide the participants into decline (2 or more points worsening) and non-decline (all other participants) groups, with 16 (18.2%) and 72 (81.8%) participants in each group, respectively. A logistic regression model showed that the odds ratio (OR) for cognitive decline was lower in participants whose responses were "engaged in hobbies" (OR = 0.07, $p$ = 0.015), "worked on a worksheet about cognitive training provided by the town hall" (OR = 0.19, $p$ = 0.026), and "had conversations over the phone" (OR = 0.28, $p$ = 0.0495). There was a significant improvement in TDAS scores after class was resumed ($p$ < 0.01). A proactive approach to intellectual activities and social ties may be important for the prevention of cognitive decline during periods of restrictions due to COVID-19. We found that cognitive function test scores before class suspension significantly improved after resuming classes. We speculate that continued participation in this class led to positive behavioral changes in daily life during periods of restriction due to COVID-19.

ethics committee (me-rinshoukenkyu@ml.adm.tottori-u.ac.jp) on request. However, access to the data sets will only be granted upon approval by all authors, the data provider (the Health measures section, Houki town), and the ethics committee.

**Funding:** The authors received no specific funding for this work.

**Competing interests:** MK, SF, and KM have no conflicts of interest to declare. KU owns a patent on the Touch Panel-type Dementia Assessment Scale and receives royalties from the Nihon Kohden Corporation (Tokyo, Japan). This does not alter our adherence to PLOS ONE policies on sharing data and materials.

## Introduction

Recently, local governments throughout Japan have made efforts against dementia. In Houki town (Tottori Prefecture, Japan), classes of about 10–20 people per class are held for community-dwelling older adults with suspected mild cognitive decline to provide cognitive decline prevention programs that combine physical exercise, cognitive training, and education on dementia and lifestyle habits [1]. A previous study has shown that combined cognitive and physical training, delivered either simultaneously or sequentially, is effective in promoting both cognitive and physical health in old age [2]. In addition, social isolation is a risk factor for poor cognitive function and dementia [3–7]. By participating in the classes in Houki town, participants interact with others; thus, they experience an increase in interpersonal communication in addition to the physical exercise and cognitive training. This may contribute synergistically to improve cognitive function. However, due to the coronavirus disease 2019 (COVID-19) pandemic, there were concerns about gathering of people causing infection transmission. The first case of COVID-19 in Japan was reported by the Ministry of Health, Labour and Welfare on January 16, 2020 [8], and COVID-19 has since spread nationwide. A state of emergency was announced by the Japanese government on April 7, 2020, and the government asked people to refrain from going out unless necessary. The Houki town hall staff has considered measures to prevent the spread of COVID-19 before the announcement of the state of emergency and has decided to suspend the class designed to help prevent cognitive decline from early March 2020. There were concerns around the negative effects of this class suspension such as deterioration of physical and cognitive health due to lack of physical exercise or cognitive training during this class, decreased social contact due to the lack of places to gather, and reduced communication. However, older patients with COVID-19 are more likely to exhibit higher mortality rates and more severe symptoms [9–14]. Furthermore, given the social situation, suspension of the class was an unavoidable decision. Although these classes resumed in June 2020, there were concerns around the progression of cognitive impairment that may have occurred during the suspension period. Therefore, when these classes resumed, participants were assessed by the Houki town hall staff using a cognitive function test and a questionnaire on lifestyle and thoughts during the suspension. They grasped the condition of the class participants and provided appropriate support.

There are ongoing concerns about further additional measures that will require restrictions due to COVID-19. It is also possible that the services provided by local governments will be suspended again in the future. Therefore, we thought it was important to determine what can be done to prevent cognitive decline. A previous study has shown that a multidomain lifestyle intervention was able to improve cognition in a group experiencing cognitive decline [15]. Other studies reported on the relationships among cognitive function and exercise, cognitive training, and social contact [2–7]. In this study, by focusing on lifestyle and awareness of activities, the purpose was to investigate the changes in cognitive function during COVID-19 restrictions. We assessed cognitive function, lifestyle, and thoughts during the class suspension in community-dwelling older adults with mild cognitive decline who participated in the class for preventing cognitive decline. We also compared cognitive function before and after the class suspension to observe changes during the period of self-restraint.

## Methods

### Study design

This was a retrospective case-control study. Data were collected in September 2020 from information stored at Houki town hall. The ethics committee of Tottori University Faculty of

Medicine (Yonago, Japan) approved the study (number: 20A076) and all procedures complied with the Declaration of Helsinki. The need for consent was waived by the ethics committee of Tottori University Faculty of Medicine because this was an observational study that used pre-collected information. The study was considered neither invasive nor intervening, and an opt-out approach was employed. All information pertaining to the study was disclosed in advance on the homepage of the Tottori University website and the notice board of Houki town hall to ensure that participants were given the opportunity to decline participation.

## Study participants

The participants were community-dwelling older adults in Houki town (Tottori Prefecture, Japan), who were enrolled in classes for prevention of cognitive decline in 2019 and 2020. Participants in this class were evaluated with suspected cognitive decline previously using a cognitive function test on a touch-pad computer (Touch Panel-type Dementia Assessment Scale: TDAS, Nihon Kohden Corporation, Tokyo, Japan [16]). There are three types of classes: a weekly class (held for four months) for new people, and a once or twice a month follow-up class (held throughout the year) for people who have completed the weekly class. Participants are evaluated for cognitive function using the TDAS once a year between January and March. Generally, the classification of the two follow-up classes is based on the degree of cognitive impairment by evaluating the TDAS scores. People with mild or greater cognitive dysfunction are invited to participate in the twice-a-month class, whereas those with a normal cognition range or extremely mild cognitive dysfunction are invited to participate in the once-a-month class. However, these classes were suspended from early March 2020 to end of May 2020 to prevent the spread of COVID-19. When the classes resumed in June 2020, participants were assessed by the Houki town hall staff using a cognitive function test and a questionnaire on lifestyle and thoughts during the suspension.

In this study, we collected data from 88 participants (mean age = 81.0 [standard deviation = 6.5] years) who met the following inclusion criteria: (1) completed the TDAS evaluation between January 2020 and March 2020 and (2) completed the TDAS evaluation and a questionnaire on lifestyle and thoughts during the class suspension between June 2020 and July 2020. The exclusion criteria comprised participants who declined the use of their data after seeing research information on the homepage of the Tottori University website and the notice board of Houki town hall. However, none of the participants declined the use of their data.

## Data collection

We collected data on the demographic characteristics, results of the cognitive function tests and the questionnaire on lifestyle and thoughts during the class suspension, and the type of class attended (once or twice a month follow-up class or a weekly class).

TDAS [16] is a cognitive function test and a modified version of the Alzheimer's Disease Assessment Scale-Cognitive Subscale (ADAS-Cog) [17], in which participants enter their answers directly onto a touch-panel computer according to the following instructions. The nine examination items include "word recognition," "following simple commands," "visual-spatial perception," "accuracy of the order of a process," "naming fingers," "orientation," "money calculation," "object recognition," and "clock time recognition." Scores ranged from 0 (all correct answers) to 101 points (all incorrect answers), so that higher scores indicate greater symptom severity.

The questionnaire was created independently by the Houki town hall staff and included questions about participants' lifestyle and thoughts during the class suspension (S1 Table), such as anxiety (worrying about physical condition, getting sick, progression of forgetfulness,

weakening of lower body, reduced food intake, decreased conversation, difficulty sleeping, and increased stress), awareness (exercising, working on cognitive training, having conversations, eating a well-balanced diet, living a well-regulated lifestyle, engaging in hobbies, and collecting information on dementia), working on cognitive training paper worksheets of the contents covered in the class provided by the town hall (sent by post once each in April and May 2020), opportunities for conversation, and method of conversation (directly, by phone, or using an online communication tool). Although other questions were present, the results of the items that were similar to the above questions are not shown in this main text. The results are shown in S2 Table. Moreover, descriptive answers were also excluded from the analysis. The questionnaire was self-administered.

## Statistical analysis

Statistical analyses were performed using SPSS (version 25, IBM Corporation, Tokyo, Japan) and EZR (version 1.41, Saitama Medical Center, Jichi Medical University, Saitama, Japan), which is a graphical user interface for R (The R Foundation for Statistical Computing, Vienna, Austria) [18]. The Shapiro-Wilk test was used to assess the normality of the data, and Levene's test was used to assess the equality of variance.

In our previous study [1] that evaluated the cognitive function of older adults with suspected cognitive decline using TDAS, we observed an average natural deterioration of 0.74 points (standard error 0.43) after a six month observational period. Based on this result, we set the criteria for cognitive decline as a 2-point or worse in the TDAS score from the January to March 2020 assessment to the June to July 2020 assessment. The results of the questionnaire were compared by dividing participants into two groups: those whose TDAS score deteriorated (decline group) and those whose TDAS score did not deteriorate (non-decline group). Differences in demographic characteristics between groups were assessed using Student's t-test or Fisher's exact test, as appropriate. Univariate and multivariate binary logistic regression analyses were used to assess associations between each of the questionnaire's results (the independent variables) and cognitive decline (the dependent variable). Because of fewer participants in the decline group, we needed to consider that the validity of the logistic model becomes problematic when the ratio of the numbers of events per variable analyzed is small [19]. Therefore, with reference to a previous study [20], for multivariable modeling, we used propensity score adjustment for age, sex, TDAS score before the class suspension, and type of class attended. The propensity score was estimated by a multivariate binary logistic regression analysis with the answers to each item in the questionnaire as the dependent variable, and the above adjustment factor as the independent variable. In the logistic regression analysis, unadjusted and adjusted odds ratios (OR) and 95% confidence intervals (CI) were calculated.

Change in the TDAS score between before and after the class suspension was evaluated using the Wilcoxon signed-rank test. We also carried out the above analyses based on the following stratifications: (1) level of cognitive function (based on January to March 2020 scores), which was based on previous studies [21, 22] that showed that the average or median TDAS score for mild cognitive impairment (MCI) was around 7 points, those with suspected mild or greater cognitive decline had a TDAS score of 7 points or higher, and those with a normal cognition range or extremely mild cognitive decline had a TDAS score of 6 points or lower; (2) type of class attended (once or twice a month); (3) level of anxiety, calculated by the total number of positive responses to items relating to "anxiety during class suspension (question 1 in S1 Table)" in the questionnaire and grouped by the median; (4) level of awareness, calculated by the total number of positive responses to items relating to "awareness during classroom suspension (question 2 in S1 Table)" in the questionnaire and grouped by the median.

All statistical significance tests were two-sided and used an alpha level of 0.05 as statistically significant.

## Results

### Participant characteristics

This study enrolled 88 participants. None of the individuals participated in the weekly class, which is only attended by new participants, and thus our subjects comprised those who participated in the follow-up classes that were held once or twice a month. All participants attended the follow-up classes in both 2019 and 2020. The participants' characteristics are summarized in Table 1. There were 26 men and 62 women, and the mean age was 81.0 (standard deviation [SD] 6.5) years. The mean TDAS score before the class suspension was 7.1 (SD 6.0), and the average degree of cognitive decline was interpreted as mild. The average time intercurring between the first and second evaluation of TDAS was 139.8 (SD 11.2) days. More than

**Table 1. Participants' characteristics.**

|  |  | All participants (n = 88) |
|---|---|---|
| Age |  | 81.0 ± 6.5 |
| Sex (male:female) |  | 26 (29.5):62 (70.5) |
| TDAS score before class suspension |  | 7.1 ± 6.0 |
| Anxiety during class suspension | Worrying about physical condition | 41 (46.6) |
|  | Getting sick | 10 (11.4) |
|  | Progressing of forgetfulness | 28 (31.8) |
|  | Weakening of lower body | 37 (42.0) |
|  | Reduced food intake | 15 (17.0) |
|  | Decreased conversation | 44 (50.0) |
|  | Difficulty sleeping | 13 (14.8) |
|  | Increased stress | 18 (20.5) |
| Awareness during class suspension | Exercising | 53 (60.2) |
|  | Working on cognitive training | 32 (36.4) |
|  | Having conversations | 37 (42.0) |
|  | Eating a well-balanced diet | 43 (48.9) |
|  | Living a well-regulated lifestyle | 47 (53.4) |
|  | Engaging in hobbies | 36 (40.9) |
|  | Collecting information on dementia | 22 (25.0) |
| Worksheet provided by the town hall [a] | Not working | 17 (20.2) |
|  | Working | 67 (79.8) |
| Opportunity for conversation [b] | Rarely | 1 (1.2) |
|  | Once a week | 3 (3.6) |
|  | Once every 2–3 days | 10 (12.0) |
|  | Almost every day | 69 (83.1) |
| Conversation method | Directly | 72 (81.8) |
|  | Using a phone | 44 (50.0) |
|  | Using an online tool | 3 (3.4) |

Data presented as mean ± standard deviation or number (%).

[a] There were 4 non-responders

[b] There were 5 non-responders

TDAS, Touch Panel-type Dementia Assessment Scale.

approximately 80% of respondents answered that they worked on the worksheet about cognitive training that was provided by the town hall, and had conversations almost every day, including direct conversations. Less than 5% of respondents answered that they had opportunities for conversation less than once a week or had conversations using an online communication medium.

## Questionnaire

The results of the questionnaire are shown in Table 2 and S2 Table. In all, 16 participants (18.2%) had cognitive decline, and 72 (81.8%) had no cognitive decline. We observed no significant differences in the age and sex ratio ($p = 0.475$, $p = 0.226$, respectively). A logistic regression model showed that the ORs for cognitive decline were lower in participants whose responses were "engaged in hobbies" (OR = 0.07, 95% CI = 0.01–0.60, $p = 0.015$), "worked on worksheet about cognitive training provided by the town hall" (OR = 0.19, 95% CI = 0.04–0.82, $p = 0.026$), and "had conversations using a phone" (OR = 0.28, 95% CI = 0.08–0.997, $p = 0.0495$).

## Cognitive function test

Fig 1A–1E show the TDAS scores before and after the class suspension. In the analysis of all participants, a significant improvement in TDAS score was observed after the class resumed (Fig 1A, $p < 0.01$). When grouping according to the degree of cognitive function before the class suspension, a significant improvement in cognitive function was observed in the group with mild or greater cognitive decline (TDAS of 7 points or more before the class suspension) (Fig 1B, $p < 0.01$). In addition, when groups were classified according to the type of class attended, a significant improvement in cognitive function was observed in the group that attended classes twice a month (Fig 1C, $p < 0.01$). Furthermore, when grouped according to level of anxiety during class suspension (low anxiety: 0–2 positive responses; high anxiety: 3–8 positive responses), there was significant improvement in cognitive function in both groups (Fig 1D, both $p < 0.05$). When participants were grouped according to level of awareness (high awareness: 3–7 positive responses; low awareness: 0–2 positive responses), a significant improvement in cognitive function was observed in the high awareness group (Fig 1E, $p < 0.01$).

## Discussion

In this study, we performed an analysis of lifestyle and awareness during the suspension of cognitive decline prevention classes due to COVID-19. We found that intellectual activities, such as "engaging in hobbies," "working on the worksheet provided by the town hall," and "having conversations on the phone" play an important role in preventing cognitive decline during a period of self-restraint. In addition, participants who had high levels of awareness during the class suspension showed significant improvement in TDAS score. Previous studies have reported that certain lifestyle activities may play an important role in the reversal of MCI to normal cognition [23]. Moreover, there is evidence of a significant association between high frequency of stimulating leisure activities in the past and reduced risk of dementia [24, 25]. Therefore, we suggest that the activities of daily life are important in preventing deterioration of cognitive function. In addition, previous studies have reported that having few social ties increased the risk for cognitive decline among elderly people [3, 26]; therefore, it is presumed that the phone was a useful communication medium in situations where the frequency of face-to-face meetings decreased due to the COVID-19 pandemic. Participants who had high awareness and engaged in many activities showed improvement in cognitive function, which

**Table 2. Comparison of questionnaire results between the cognitive decline and non-decline group.**

| | Decline group [a] | Non-decline group [b] | Unadjusted | | Adjusted | |
|---|---|---|---|---|---|---|
| | (n = 16) | (n = 72) | OR (95% CI) | P value | OR (95% CI) | P value |
| Age | 82.1 ± 7.1 | 80.8 ± 6.3 | | | | |
| Sex (male:female) | 7 (43.8): 9 (56.3) | 19 (26.4): 53 (73.6) | | | | |
| Anxiety during class suspension | | | | | | |
| Worrying about physical condition | 8 (50.0) | 33 (45.8) | 1.18 (0.40–3.49) | 0.763 | 1.40 (0.45–4.36) | 0.556 |
| Getting sick | 1 (6.3) | 9 (12.5) | 0.47 (0.05–3.97) | 0.485 | 0.45 (0.05–3.88) | 0.465 |
| Progressing of forgetfulness | 4 (25.0) | 24 (33.3) | 0.67 (0.19–2.29) | 0.519 | 0.67 (0.19–2.32) | 0.524 |
| Weakening of lower body | 9 (56.3) | 28 (38.9) | 2.02 (0.68–6.04) | 0.208 | 2.12 (0.69–6.49) | 0.188 |
| Reduced food intake | 4 (25.0) | 11 (15.3) | 1.85 (0.50–6.79) | 0.355 | 1.78 (0.48–6.62) | 0.386 |
| Decreased conversation | 9 (56.3) | 35 (48.6) | 1.36 (0.46–4.04) | 0.581 | 1.51 (0.48–4.74) | 0.477 |
| Difficulty sleeping | 2 (12.5) | 11 (15.3) | 0.79 (0.16–3.98) | 0.777 | 0.81 (0.16–4.12) | 0.798 |
| Increased stress | 5 (31.3) | 13 (18.1) | 2.06 (0.61–6.96) | 0.243 | 2.10 (0.59–7.47) | 0.250 |
| Awareness during class suspension | | | | | | |
| Exercising | 8 (50.0) | 45 (62.5) | 0.60 (0.20–1.78) | 0.358 | 0.62 (0.20–1.92) | 0.407 |
| Working on cognitive training | 6 (37.5) | 26 (36.1) | 1.06 (0.35–3.26) | 0.917 | 0.86 (0.26–2.84) | 0.804 |
| Having conversations | 5 (31.3) | 32 (44.4) | 0.57 (0.18–1.80) | 0.337 | 0.59 (0.18–1.91) | 0.378 |
| Eating a well-balanced diet | 7 (43.8) | 36 (50.0) | 0.78 (0.26–2.31) | 0.651 | 0.82 (0.26–2.60) | 0.742 |
| Living a well-regulated lifestyle | 11 (68.8) | 36 (50.0) | 2.20 (0.69–6.97) | 0.180 | 2.41 (0.74–7.84) | 0.145 |
| Engaging in hobbies | 1 (6.3) | 35 (48.6) | 0.07 (0.01–0.56) | 0.012 | 0.07 (0.01–0.60) | 0.015 |
| Collecting information on dementia | 4 (25.0) | 18 (25.0) | 1.00 (0.29–3.49) | 1.000 | 0.85 (0.23–3.09) | 0.798 |
| Worksheet provided by the town hall [c] | | | | | | |
| Not working | 6 (42.9) | 11 (15.7) | 1 (reference) | | 1 (reference) | |
| Working | 8 (57.1) | 59 (84.3) | 0.25 (0.07–0.86) | 0.028 | 0.19 (0.04–0.82) | 0.026 |
| Opportunity for conversation [d] | | | | | | |
| Rarely | 0 (0) | 1 (1.4) | 1 (reference) [e] | | 1 (reference) [e] | |
| Once a week | 0 (0) | 3 (4.3) | | | | |
| Once every 2–3 days | 1 (7.7) | 9 (12.9) | | | | |
| Almost every day | 12 (92.3) | 57 (81.4) | 2.74 (0.33–23.0) | 0.354 | 3.84 (0.41–35.80) | 0.238 |
| Conversation method | | | | | | |
| Directly | 12 (75.0) | 60 (83.3) | 0.60 (0.17–2.18) | 0.438 | 0.64 (0.16–2.49) | 0.514 |
| Using a phone | 4 (25.0) | 40 (55.6) | 0.27 (0.08–0.91) | 0.034 | 0.28 (0.08–0.997) | 0.0495 |
| Using an online tool | 0 (0) | 3 (4.2) | – [f] | | – [f] | |

Data presented as mean ± standard deviation or number (%). Propensity score was used as an adjustment covariate.

[a] There was a deterioration of 2 points or more in the TDAS score after resuming classes compared to that before class suspension.

[b] There was no deterioration of 2 points or more in the TDAS score after resuming classes compared to that before class suspension.

[c] There were 2 non-responders in the decline group, and 2 non-responders in the non-decline group

[d] There were 3 non-responders in the decline group, and 2 non-responders in the non-decline group

[e] Reference was the total value of "almost none," "once a week," and "once every 2–3 days"

[f] Analysis was not possible because there were no participants in the decline group that had a conversation using an online communication tool

OR, Odds ratio; CI, Confidence interval; TDAS, Touch Panel-type Dementia Assessment Scale.

demonstrates that it is beneficial to continue any possible activities while refraining from activities due to COVID-19.

Previous studies have shown that in older adults, the spread of COVID-19 may lead to decreased social [27] and physical activity time [28], and an increasing risk of problematic mental health symptoms such as depression and anxiety due to more days spent staying at home [29]. Physical inactivity, social isolation, and depression in later life (older than 65 years)

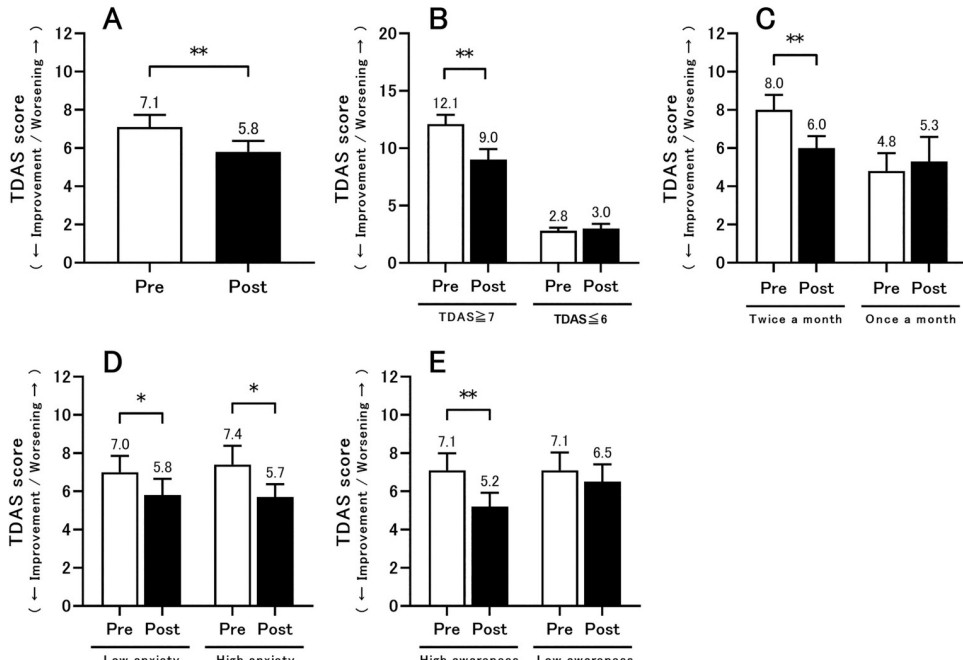

**Fig 1. Results of TDAS before class suspension (pre) and after class resumption (post).** (A) The results of all participants are shown (n = 88). (B) The results of the mild or greater cognitive decline group (TDAS score of 7 points or more) before class suspension (n = 41 [13 men and 28 women], mean age = 82.3 [SD = 6.2] years), and the normal cognition or extremely mild cognitive decline group (TDAS score of 6 points or less) before class suspension (n = 47 [13 men and 34 women], mean age = 79.9 [SD = 6.5] years). (C) The results of subjects who participated in the class held twice a month (n = 64 [19 men and 45 women], mean age = 81.9 [SD = 6.6] years) and once a month (n = 24 [7 men and 17 women], mean age = 78.6 [SD = 5.5] years). (D) The results of low anxiety participants (n = 53 [16 men and 37 women], mean age = 80.8 [SD = 6.8] years) and high anxiety participants (n = 35 [10 men and 25 women], mean age = 81.3 [SD = 5.9] years). (E) The results of high awareness participants (n = 48 [10 men and 38 women], mean age = 81.2 [SD = 6.5] years) and low awareness participants (n = 40 [16 men and 24 women]), mean age = 80.7 [SD = 6.5] years). All data indicate mean ± standard error, and the number on the error bar is the mean value. *p<0.05, **p<0.01. TDAS, Touch Panel-type Dementia Assessment Scale; SD, standard deviation.

are risk factors for dementia [7]. Moreover, in a study investigating the effects of COVID-19 lockdown in Italy, telephone interviews with caregivers of 31 subjects with MCI revealed a worsening of cognition in 41.9% of people [30]. Although Japan did not impose any lockdown, there was concern that class suspension and staying at home would lead to cognitive decline. However, although 31.8% of the participants answered in the questionnaire that they were worried about the progress of forgetfulness, 18.2% of the participants in our study had cognitive decline. Moreover, cognitive function improved after the classes resumed. Anxiety symptoms have also been reported to increase the risk of developing MCI and dementia [31]; however, cognitive function improved after class resumed, regardless of the levels of anxiety. In addition, subjects who participated in the class held twice a month and subjects with mild or more severe cognitive decline were concerned about their decline in cognitive function due to class suspension. However, no adverse effect on cognitive function was observed because of changes in the environment due to COVID-19. One possible reason can be that the extent of the measures taken to counteract the COVID-19 pandemic in Japan was on a voluntary basis and presumably irrelevant in determining a significant detrimental effect on the cognitive functions of Japanese elderly with mild cognitive decline. Another reason can be that continued attendance of the classes until the suspension may explain the limited cognitive decline observed in our study. Although cultural differences and difference in survey years must be

taken into consideration, it has been proposed that both in Japan and other countries, older adults' knowledge of dementia and cognitive disorders is currently deficient and that public health approaches such as educational interventions or national campaigns are necessary [32–35]. Although we did not investigate the degree of knowledge in our participants, we can infer that the participants in this study had some understanding of the type of lifestyle that is important to prevent cognitive decline because they had already been participating in the classes. Therefore, we suspect that many participants carried out beneficial actions to prevent the deterioration of cognitive function during the period of self-restraint imposed due to COVID-19.

We found that few people had conversations using an online communication medium. According to the results of the 2019 Communications Usage Trend Survey in Japan released by the Ministry of Internal Affairs and Communications, the usage status of social networking services was reported to be 40.7% for people aged 70–79 and 42.8% for people aged 80 and over [36]. In a study examining the compliance rate of computer-based cognitive training in older adults with increased risk of dementia, 37% did no train at all, and previous experience with computers, being married or cohabiting, better memory performance, and positive expectations toward the study predicted greater likelihood for starting the training [37]. In addition, a study investigating the frequency of exposure to health information from information and communication technologies reported that older adult respondents were less frequently exposed to health information from websites and social media sites [38]. Taken together, we consider it important to provide support for older adults who are suspected of having cognitive decline by using more traditional methods, such as telephone and print media, rather than digital mediums, especially during emergency situations. Furthermore, 79.8% of participants answered that during the class suspension, they worked on the cognitive training worksheet provided by the town hall. Although individual awareness was an important factor that influenced whether an individual worked on the worksheet provided by the town hall, the results suggested that it is important to provide information in an appropriate manner such as through paper.

Finally, this study has several limitations. First, the subjects of this study were community-dwelling older adults, who participated in a class for the prevention of cognitive decline. We did not examine older adults who do not participate in such classes. Therefore, we may not be able to generalize these results to those older adults who experienced restrictions due to COVID-19 but do not participate in such programs. The second limitation is that we cannot control for the carryover effect of attending the classes. In a previous study, it was suggested that the effect of the cognitive decline prevention program used in the class may not be sustained for up to 6 months after the intervention is stopped [1]. However, in this study, the class was only suspended for around three months. Therefore, although we observed improvements in cognitive function after the class suspension, it is possible that participation in the class before the suspension may have had some influence. The third limitation was the small sample size of participants with cognitive decline (n = 16), which limited the accuracy of our data. The fourth research limitation is that we did not consider all the possible confounding factors that may affect cognitive function. For example, the status of infection of COVID-19 in participants may also have an effect on cognitive performance [39, 40] and daily activities. As the data available were limited due to retrospective study, we could only adjust for a few factors. However, since the number of patients with COVID-19 in Tottori Prefecture in Japan was three from March to May 2020 and the classes resumed without any problems, it is estimated that no participants were infected with COVID-19 during the class suspension. The fifth limitation is that the subjects of this study do not include those who did not come to the class when it resumed. The target sample size was set at about 120 participants at the research planning stage, which was the total number of class participants in 2019 and 2020. Participation in the class is

voluntary, so there may be various reasons for absence (e.g., having another plan, concerned about COVID-19 infection, etc.).

## Conclusions

A proactive approach toward intellectual activities and social ties may be important in preventing cognitive decline during periods of self-restraint due to COVID-19. We observed significant improvement in cognitive function scores following the resumption of the class. We speculate that participants' understanding of the importance of an active lifestyle (reflected in their continued attendance of the classes) may explain these improvements. We propose that the class is important not only to prevent cognitive decline, but also to prepare for cognitive decline in situations of restrictive living, such as during the COVID-19 pandemic.

## Supporting information

**S1 Table. English translation of the relevant questions used in this study.**
(DOCX)

**S2 Table. Comparison of questionnaire results between the cognitive decline and non-decline group (results not shown the main text).**
(DOCX)

## Acknowledgments

We thank the staff of the Health measures section, Houki town who cooperated with us during this study. In addition, we would like to thank Editage (www.editage.com) for English language editing.

## Author Contributions

**Conceptualization:** Minoru Kouzuki, Katsuya Urakami.

**Data curation:** Minoru Kouzuki.

**Formal analysis:** Minoru Kouzuki, Shota Furukawa, Keisuke Mitani.

**Supervision:** Katsuya Urakami.

**Writing – original draft:** Minoru Kouzuki.

**Writing – review & editing:** Minoru Kouzuki, Katsuya Urakami.

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
