## [Decision Letter · Decision Letter 0]

1 Sep 2021

PONE-D-21-06873

Impact of the COVID-19 pandemic on cognitive function in Japanese community-dwelling older adults in a class for preventing cognitive decline

PLOS ONE

Dear Dr. Kouzuki,

Thank you for submitting your manuscript to PLOS ONE. After careful consideration, we feel that it has merit but does not fully meet PLOS ONE’s publication criteria as it currently stands. Therefore, we invite you to submit a revised version of the manuscript that addresses the points raised during the review process.

We look forward to receiving your revised manuscript.

Kind regards,

Ghulam Md Ashraf, Ph.D.

Academic Editor

PLOS ONE

Journal Requirements:

2. For this observational study, please avoid causal-sounding language (such as 'impact' or 'effect') when reporting associations, including in the title.

3. Thank you for stating the following in the Competing Interests/Financial Disclosure* (delete as necessary) section:

“I have read the journal's policy and the authors of this manuscript have the following competing interests: MK, SF, and KM have no conflicts of interest to declare. KU owns a patent on the Touch Panel-type Dementia Assessment Scale and receives royalties from Nihon Kohden Corporation (Tokyo, Japan).”

We note that one or more of the authors are employed by a commercial company: Touch Panel-type Dementia Assessment Scale & Nihon Kohden Corporation (Tokyo, Japan)

Reviewers' comments:

Reviewer's Responses to Questions

**Comments to the Author**

1. Is the manuscript technically sound, and do the data support the conclusions?

Reviewer #1: Yes

Reviewer #2: Partly

2. Has the statistical analysis been performed appropriately and rigorously? 

Reviewer #1: Yes

Reviewer #2: I Don't Know

3. Have the authors made all data underlying the findings in their manuscript fully available?

Reviewer #1: Yes

Reviewer #2: Yes

4. Is the manuscript presented in an intelligible fashion and written in standard English?

Reviewer #1: Yes

Reviewer #2: Yes

5. Review Comments to the Author

Reviewer #1: This is a fairly well written manuscript. The COVID pandemic is pretty recent and any study performed with reference to the same will add value to the knowledge. The authors of current manuscript did the same. They investigated the effect of suspension of classes that were being held for elderly individuals for preventing cognitive decline.

I have only minor issues-

1. The opening statement of the manuscript, the first line of the manuscript reads incorrect. Use of too many 'ands' in the statement. Also, what is 'thoughts on cognitive function'? Kindly modify the statement, it reads very vague.

2. Authors have not mentioned age of the individuals in the abstract and age and number of patients recruited in th MM section. They only mention it finally in the result section. KIndly add this information in other two sections.

3. Format of Table-1 can be improved. Kindly make it concise.

Reviewer #2: This title seems better "Impact of the COVID19 pandemic on cognitive function of Japanese community-dwelling older adults in a class for preventing cognitive decline".

In lines 43-44, “Participation in the class leads to the formation

of a community”. It would be helpful to specify or briefly mention/explain the class structure.

Also in line 52> specify or briefly mention/explain the class structure

At many places In the intro and other sections “the class(es)” can be changed to ‘this(these) class(es)’’ after having briefly mentioned what those classes are at the beginning.

In line 86, delete “not”.

I didn’t not understand this line (112-113) “The exclusion criteria were subjects who declined the use of their data”.

From Table 1 & 2, I couldn’t get the clear idea as to what the observed cognitive parameters were before and/or after the class suspension.

I am also concerned about the small sample size of cognitive decline group. I was also wondering if three months (early March to end May) is good enough time window to assess the dementia related symptoms.

6. PLOS authors have the option to publish the peer review history of their article (what does this mean?). If published, this will include your full peer review and any attached files.

Reviewer #1: No

Reviewer #2: **Yes: **Zeeshan Banday

---

## [Author Response · Author response to Decision Letter 0]

8 Sep 2021

Responses to the Journal Requirements

Response: Thank you for this reminder. We have checked PLOS ONE's style requirements. We made some corrections in the arrangement of contents within a cell in Tables 1, 2, and S2.

2. For this observational study, please avoid causal-sounding language (such as 'impact' or 'effect') when reporting associations, including in the title.

Response: As you suggested, we have revised the title and some parts of the text as much as possible.

3. Thank you for stating the following in the Competing Interests/Financial Disclosure* (delete as necessary) section:

“I have read the journal's policy and the authors of this manuscript have the following competing interests: MK, SF, and KM have no conflicts of interest to declare. KU owns a patent on the Touch Panel-type Dementia Assessment Scale and receives royalties from Nihon Kohden Corporation (Tokyo, Japan).”

We note that one or more of the authors are employed by a commercial company: Touch Panel-type Dementia Assessment Scale & Nihon Kohden Corporation (Tokyo, Japan)

Response: The Touch Panel-type Dementia Assessment Scale (TDAS) is the name of the cognitive function test, and Nihon Kohden Corporation is the company that sells TDAS. KU receives royalties from Nihon Kohden Corporation, but is not employed. In addition, this study did not use any royalties and was not funded by Nihon Kohden Corporation. This study was independently planned and carried out by researchers who were not employed by a commercial company, and Nihon Kohden Corporation had no role in the study design, data collection and analysis, decision to publish, or preparation of the manuscript. Therefore, there are no changes to the Funding Statement as the following: “The authors received no specific funding for this work.”

Response: Within our Competing Interests Statement, we added, "This does not alter our adherence to PLOS ONE policies on sharing data and materials.” Please refer to the following:

“MK, SF, and KM have no conflicts of interest to declare. KU owns a patent on the Touch Panel-type Dementia Assessment Scale and receives royalties from the Nihon Kohden Corporation (Tokyo, Japan). This does not alter our adherence to PLOS ONE policies on sharing data and materials.”

Response: We checked the reference list and no retracted articles were included. However, some errors in the reference list were revised.

 

Responses to the comments of Reviewer 1

1. The opening statement of the manuscript, the first line of the manuscript reads incorrect. Use of too many 'ands' in the statement. Also, what is 'thoughts on cognitive function'? Kindly modify the statement, it reads very vague.

Response: We apologize for the lack of clarity. We have revised the text in the opening part of the Abstract.

2. Authors have not mentioned age of the individuals in the abstract and age and number of patients recruited in the MM section. They only mention it finally in the result section. Kindly add this information in other two sections.

Response: We appreciate this valuable suggestion. We have added information on the age of the individuals in the Abstract, as well as the age and number of participants recruited in the Methods/Study participants section. In addition, we have revised the Results/Participant characteristics section to avoid duplication of text.

3. Format of Table-1 can be improved. Kindly make it concise.

Response: Thank you for your comment. We have tried to make the format of Table 1 as concise as possible. In addition, we have changed the sex description format in Table 2.

Responses to the comments of Reviewer 2

1. This title seems better "Impact of the COVID19 pandemic on cognitive function of Japanese community-dwelling older adults in a class for preventing cognitive decline".

Response: Thank you for your comment. However, the Journal Requirements pointed out to avoid causal-sounding language (such as 'impact' or 'effect') when reporting associations. Therefore, we considered the following title and short title as alternatives:

Title: Examination of the cognitive function of Japanese community-dwelling older adults in a class for preventing cognitive decline during the COVID-19 pandemic

Short title: Examination of cognitive function of Japanese older adults during the COVID-19 pandemic

2. In lines 43-44, “Participation in the class leads to the formation of a community”. It would be helpful to specify or briefly mention/explain the class structure. Also in line 52> specify or briefly mention/explain the class structure.

Response: We appreciate this valuable suggestion. We have changed the text related to the class to improve its coherence. Specifically, we added a description of the class structure on line 38 and corrected the text on lines 44 and 54.

3. At many places In the intro and other sections “the class(es)” can be changed to ‘this(these) class(es)’’ after having briefly mentioned what those classes are at the beginning.

Response: We agree with your recommendation. We already explained the class at the beginning of the introduction and Methods/Study participants; thus, we changed some from “the class(es)” to “this(these) class(es)” as advised.

4. In line 86, delete “not”.

Response: Thank you for this comment. We have revised the sentences accordingly in line 89.

5. I didn’t not understand this line (112-113) “The exclusion criteria were subjects who declined the use of their data”.

Response: We apologize for the lack of clarity. In this study, all information pertaining to the study was disclosed in advance on the homepage of the Tottori University website and the notice board of the Houki town hall to ensure that participants were given the opportunity to decline participation (opt-out approach). Therefore, the exclusion criteria meant that participants who declined the use of their data were excluded. This was the item that we described in the application form for ethics review (the research proposal); thus, we have included it in the text as well. We have added an explanation of the exclusion criteria to improve its coherence.

6. From Table 1 & 2, I couldn’t get the clear idea as to what the observed cognitive parameters were before and/or after the class suspension.

Response: As we wrote in the Methods section, cognitive function was evaluated using only TDAS. However, we found the description difficult to understand; thus, we have revised the text of the Methods/Data Collection and added annotations for the decline and non-decline groups in Table 2.

7. I am also concerned about the small sample size of cognitive decline group. I was also wondering if three months (early March to end May) is good enough time window to assess the dementia related symptoms.

Response: Thank you for this very important comment. As the reviewer has pointed out, we also considered the small sample size of the cognitive decline group as a limitation of our study (described as the third limitation in the Discussion section). Unfortunately, we were unable to collect any additional data. We have added a little description about the sample size; thus, the reader should pay attention to the interpretation of the results. Regarding the time window, three months may be a short period, but because daily life has changed significantly due to COVID-19, we performed this study because we believe that even class suspension of three months would have a negative effect on cognitive function. In fact, a study has been reported that in a study investigating the effects of about two months of COVID-19 lockdown in Italy, telephone interviews with caregivers of 31 subjects with MCI revealed a worsening of cognition in 41.9% of individuals (ref 30: Baschi R, et al. Front Psychiatry. 2020; 11: 590134.). 

Corrections other than reviewer comments

Some errors in Table 2, abbreviation notation (lines 48 and 125), typographical errors (lines 156 and 287), and reference lists (lines 399, 404, 410, 428, 441, 443, 463, 467, and 511) have been corrected. However, these changes did not affect the interpretation of the results. We apologize for the lack of confirmation.

---

## [Decision Letter · Decision Letter 1]

5 Oct 2021

PONE-D-21-06873R1Examination of the cognitive function of Japanese community-dwelling older adults in a class for preventing cognitive decline during the COVID-19 pandemicPLOS ONE

Dear Dr. Kouzuki,

Thank you for submitting your manuscript to PLOS ONE. After careful consideration, we feel that it has merit but does not fully meet PLOS ONE’s publication criteria as it currently stands. Therefore, we invite you to submit a revised version of the manuscript that addresses the points raised during the review process.

We look forward to receiving your revised manuscript.

Kind regards,

Ghulam Md Ashraf, Ph.D.

Academic Editor

PLOS ONE

Journal Requirements:

Additional Editor Comments (if provided):

The authors are advised to address minor concerns raised by the reviewer.

Reviewers' comments:

Reviewer's Responses to Questions

**Comments to the Author**

1. If the authors have adequately addressed your comments raised in a previous round of review and you feel that this manuscript is now acceptable for publication, you may indicate that here to bypass the “Comments to the Author” section, enter your conflict of interest statement in the “Confidential to Editor” section, and submit your "Accept" recommendation.

Reviewer #1: All comments have been addressed

Reviewer #2: All comments have been addressed

2. Is the manuscript technically sound, and do the data support the conclusions?

Reviewer #1: Yes

Reviewer #2: Partly

3. Has the statistical analysis been performed appropriately and rigorously? 

Reviewer #1: Yes

Reviewer #2: I Don't Know

4. Have the authors made all data underlying the findings in their manuscript fully available?

Reviewer #1: Yes

Reviewer #2: Yes

5. Is the manuscript presented in an intelligible fashion and written in standard English?

Reviewer #1: Yes

Reviewer #2: Yes

6. Review Comments to the Author

Reviewer #1: (No Response)

Reviewer #2: In the revised manuscript “By participating in this class…” in line 44 now becomes incoherent. It has to be preceded with a sentence explaining what ‘this class’ actually is. Now that the authors have changed the structure of this sentence-in the process made it more complex ‘this’ in the sentence may be changed to ‘the’ OR better simplify this sentence.

In my previous review, I had asked the authors to change ‘the class(es) to this (these) class(es) at certain places giving readers a clear reference as to what those classes are, which the authors have actually done. In the revised manuscript, however, I noticed that the authors have made these changes at many other places where it was not required at all (e.g in line 65). The authors must fix such inconsistencies before the manuscript can be accepted for publication.

Lines 116-118 are contradictory.

7. PLOS authors have the option to publish the peer review history of their article (what does this mean?). If published, this will include your full peer review and any attached files.

Reviewer #1: No

Reviewer #2: **Yes: **Zeeshan Z. Banday

---

## [Author Response · Author response to Decision Letter 1]

25 Oct 2021

Responses to the Journal Requirements

Response: We checked the reference list, and no retracted articles were included. However, the details of reference 40 were revised.

Responses to the comments of Reviewer 2

In the revised manuscript “By participating in this class…” in line 44 now becomes incoherent. It has to be preceded with a sentence explaining what ‘this class’ actually is. Now that the authors have changed the structure of this sentence-in the process made it more complex ‘this’ in the sentence may be changed to ‘the’ OR better simplify this sentence.

Response: Thank you for your comment. We have revised the sentences in lines 44-47.

In my previous review, I had asked the authors to change ‘the class(es) to this (these) class(es) at certain places giving readers a clear reference as to what those classes are, which the authors have actually done. In the revised manuscript, however, I noticed that the authors have made these changes at many other places where it was not required at all (e.g in line 65). The authors must fix such inconsistencies before the manuscript can be accepted for publication.

Response: We apologize because we did not understand your intentions correctly. We changed some from “this class” to “the class” as you have pointed out. If the correction is incorrect, please let me know in detail.

Lines 116-118 are contradictory.

Response: We completely agree with you. We have revised the sentences in lines 116-118.

---

## [Decision Letter · Decision Letter 2]

29 Nov 2021

Examination of the cognitive function of Japanese community-dwelling older adults in a class for preventing cognitive decline during the COVID-19 pandemic

PONE-D-21-06873R2

Dear Dr. Kouzuki,

We’re pleased to inform you that your manuscript has been judged scientifically suitable for publication and will be formally accepted for publication once it meets all outstanding technical requirements.

Kind regards,

Ghulam Md Ashraf, Ph.D.

Academic Editor

PLOS ONE

Additional Editor Comments (optional):

Reviewers' comments:

Reviewer's Responses to Questions

**Comments to the Author**

1. If the authors have adequately addressed your comments raised in a previous round of review and you feel that this manuscript is now acceptable for publication, you may indicate that here to bypass the “Comments to the Author” section, enter your conflict of interest statement in the “Confidential to Editor” section, and submit your "Accept" recommendation.

Reviewer #2: All comments have been addressed

2. Is the manuscript technically sound, and do the data support the conclusions?

Reviewer #2: Yes

3. Has the statistical analysis been performed appropriately and rigorously? 

Reviewer #2: I Don't Know

4. Have the authors made all data underlying the findings in their manuscript fully available?

Reviewer #2: Yes

5. Is the manuscript presented in an intelligible fashion and written in standard English?

Reviewer #2: Yes

6. Review Comments to the Author

Reviewer #2: I do not have any further comments. Much of the comments have been addressed by the authors. Thanks to the authors!

7. PLOS authors have the option to publish the peer review history of their article (what does this mean?). If published, this will include your full peer review and any attached files.

Reviewer #2: **Yes: **Zeeshan Z. Banday

---

## [Editor Report · Acceptance letter]

3 Dec 2021

PONE-D-21-06873R2 

Examination of the cognitive function of Japanese community-dwelling older adults in a class for preventing cognitive decline during the COVID-19 pandemic 

Dear Dr. Kouzuki:

I'm pleased to inform you that your manuscript has been deemed suitable for publication in PLOS ONE. Congratulations! Your manuscript is now with our production department. 

Kind regards, 

on behalf of

Dr. Ghulam Md Ashraf 

Academic Editor

PLOS ONE